# Oxidation Stress-Mediated MAPK Signaling Pathway Activation Induces Neuronal Loss in the CA1 and CA3 Regions of the Hippocampus of Mice Following Chronic Cold Exposure

**DOI:** 10.3390/brainsci9100273

**Published:** 2019-10-12

**Authors:** Bin Xu, Li-Min Lang, Shuai Lian, Jing-Ru Guo, Jian-Fa Wang, Huan-Min Yang, Shi-Ze Li

**Affiliations:** College of Animal Science and Veterinary Medicine, Heilongjiang Bayi Agricultural University, Daqing 163319, China; xubin@byau.edu.cn (B.X.); langlanglimin@163.com (L.-M.L.); lianlianshuai@163.com (S.L.); byndgjr@163.com (J.-R.G.); wjflw@sina.com (J.-F.W.); byndlsz@163.com (S.-Z.L.)

**Keywords:** cold stress, MAPK signaling pathway, hippocampus oxidation stress, neuronal loss

## Abstract

Chronic stress can damage homeostasis and induce various primary diseases. Although chronic cold stress is becoming an increasing problem for people who must work or live in extreme environments, risk-induced diseases in the central nervous system remain unstudied. Male C57BL/6 mice were exposed to an environment of 4 °C, 3 h per day for 1, 2, and 3 weeks and homeostasis in the hippocampus and neuronal apoptosis were evaluated by Western blotting, immunohistochemistry, TdT-mediated dUTP Nick-End Labeling (TUNEL) staining, and immunofluorescence. The phenomena of oxidation stress, MAPK signaling pathway activation, anti-oxidation protein release, neuronal apoptosis increases, and neuronal proliferation inhibition were demonstrated in the CA1 and CA3 regions of mouse hippocampal tissues following cold exposure. We speculated that these phenomena were mediated by the MAPK pathway and were closely linked with oxidative stress in the hippocampus. This study provides novel concepts regarding neurodegenerative diseases, suggesting that chronic cold stress may be a critical factor to induce neurodegenerative diseases.

## 1. Introduction

Stress is an adaptive response of the body to internal or external altered homeostasis [1]. As we know, homeostasis is a basic state of steady internal physical and chemical conditions maintained by living systems. If the intensity of the stressor is under a certain threshold, the body may be positively affected, but if the intensity of stressor overloads the threshold level of the body [2], this could disrupt the homeostasis and induce physiological and pathological diseases, such as anxiety neurosis, depression, and endocrine dyscrasia [3]. Cold exposure is a specific stressor for humans and other organisms in some extreme environments and temperatures, and cold exposure has already been reported to be closely related with the process of energy metabolism and immunological functions [4,5], resulting in endogenous or secondary diseases [6,7]. Some studies have demonstrated that cold exposure increased the risk of neurodegenerative diseases [8,9], but the impacts on homeostasis and neurons in the brain are still unknown.

The hippocampus is an important component in the limbic system of the brain, and plays an important role in memory, learning, and behavior. The CA1 and CA3 limbic regions are both functional regions of the hippocampus that have been studied widely regarding stress responses; it would suffer serious impact by minute changes in brain homeostasis [10]. Studies have reported that the structure and function of the hippocampus suffer irreversible damage following chronic stress [11,12]. In addition, the hippocampus is a target and center of negative feedback in the hypothalamic-pituitary-adrenal (HPA) axis, and therefore, diseases relevant to abnormal activation of the HPA axis are linked to stress [13]. In our previous study, the phenomena of neuronal loss and neuroinflammation in the CA1 and CA3 hippocampus regions of mice after cold exposure were reported [14]. However, the mechanism of neuronal loss and altered homeostasis in the hippocampus is still not clear.

Oxidative stress is the imbalance resulting from excess free radicals and the inability of the body to counteract the damage caused by them, which increases the risk of aging and metabolic diseases [15]. Free radicals can destabilize homeostasis and interact with cellular DNA, proteins, and lipids [16]. However, not all reactive oxygen species are harmful; a level under the adjustable threshold of homeostasis can help eliminate the pathogen-associated molecular patterns (PAMPs), including those from pathogens and microbes [17]. Some research has reported that oxidative stress in the hippocampus, which is induced by stress, mediates and accelerates the process of apoptosis, increasing the risk of neurodegenerative diseases [18,19]. The impacts on homeostasis and hippocampal neurons caused by oxidative stress mediated by cold exposure in the hippocampus of mouse are not known. Here, we investigated the phenomena induced by oxidative stress that was mediated from cold exposure in the hippocampus of mice, and characterized the relationship between oxidative stress and cold stress in the mouse hippocampus.

## 2. Materials and Methods

### 2.1. Animals and the Cold Sress Model

Five-week-old C57BL/6 mice (males, 23–25 g) were purchased from Charles River (Beijing, China). All mice were pre-fed in a climatic chamber to adapt to the environmental temperature of 24 ± 2 °C and relative humidity of 40% under a 12/12 h light/dark cycle (light on from 8:00 a.m.to 8:00 p.m.), with free access to food and water for 1 week. Next, the mice were randomly divided into the following four groups: cold exposure for 1 week (CE1W) group, cold exposure for 2 weeks (CE2W) group, cold exposure for 3 weeks (CE3W) group, and the room temperature (RT) group (*n* = 9). The conditions involving cold exposure were based on our previous study [20]. The mice in the CE1W, CE2W, and CE3W groups were exposed to a climatic chamber at 4 °C and a humidity of 40% for 3 h/day. The mice were then returned to the original environment of 24 ± 2 °C and relative humidity of 40% under light. The mice of the RT group remained the entire time in an environment of 24 ± 2 °C and relative humidity of 40% as controls. The mice were exposed to chronic cold for 1, 2, or 3 weeks. All experimental procedures were approved by the Management Committee of the Experimental Animal Center of Heilongjiang Bayi Agricultural University.

### 2.2. Brain Tissue Collection

After the last cold exposure, all mice of each group (*n* = 3 per group) were immediately anesthetized (1% pentobarbital PBS solution) and transcardially perfused to fix the brain with ice, normal saline (NS), and 4% paraformaldehyde. The brains were then perfused and fixed in 4% paraformaldehyde for 72 h, immersed in a solution of 30% sucrose for 12–24 h, and then cut into 30-μm thick coronal sections (*n* = 10 per brain) after being snap frozen on a freezing microtome (CM1850, Leica Instrument, Wetzlar, Germany). The other hippocampus tissue used for Western blotting or malondialdehyde (MDA) analysis (*n* = 6 per group) were also isolated and stored at −80 °C until use.

### 2.3. Immunohistochemistry

As previously reported, brain sections were rinsed with PBS two times, then incubated with 0.3% H_2_O_2_ for 20 min and then blocked with 1% horse serum albumin (C0262, Beyotime, Hangzhou, China) at room temperature for 15 min, then incubated with MAP2 primary antibodies (17490-1-AP, 1:100; Proteintech, Wuhan, China) at 4 °C overnight. The sections were then rinsed with PBS for 15 min, followed by incubation with the appropriate secondary antibodies at room temperature for 1 h. The sections were then treated with Diaminobenzidine (DAB) (DA1010; Solarbio, Beijing, China), followed by an alcohol gradient to dehydrate them, and finally cleared in xylene and viewed using a laser scanning confocal microscope (TCS SP2; Leica, Wetzlar, Germany) to count the positive cells.

### 2.4. TUNEL Staining

Brain sections were rinsed two times with PBS, incubated with 0.3% H_2_O_2_ for 15 min, rinsed in PBS for 15 min, and incubated with TdT-mediated dUTP Nick-End Labeling (TUNEL) stain solution (C1086; Beyotime, Beijing, China) according to the manufacturer’s instructions. The sections were viewed and photographed with a laser scanning confocal microscope (TCS SP2; Leica, Wetzlar, Germany) and the numbers of TUNEL-positive cells were counted.

### 2.5. Western Blot Analysis

Total hippocampus proteins were extracted using 100 μL (Radio Immunoprecipitation Assay) RIPA buffer (P0013B; Beyotime, HangZhou, China) with 1% phenylmethylsulfonyl fluoride (PMSF) (ST505; Beyotime, HangZhou, China). Protein concentration was measured using the enhanced BCA protein assay kit (P0009; Beyotime, HangZhou, China), following the manufacturer’s instructions. Approximately 30 μg of total protein was separated using SDS-PAGE, and the gel was transferred onto a 0.45 μm polyvinylidene fluoride (PVDF) membrane (Millipore, Darmstadt, Germany). Membranes were blocked in TBST (Tris-HCl, NaCl, and Tween 20) with 5% nonfat milk for 1 h, then incubated with appropriate primary antibodies overnight at 4 °C. The primary antibodies used were as follows: nuclear factor-like 2 (Nrf2) (#16396-1-AP, 1:1000), Kelch-like ECH-associated protein (KEAP) 1 (#10503-2-AP, 1:1000), catalase (CAT) (#21260-1-AP, 1:1000), superoxide dismutase (SOD) (#10269-1-AP, 1:1000), heme oxygenase (HO) 1 (#10701-1-AP, 1:1000), cytochrome C (#10993-1-AP, 1:1000), B-cell lymphoma 2 (Bcl-2) (#12789-1-AP, 1:1000), Bcl-2-associated X (Bax) (#50599-2-Ig, 1:1000), caspase 3 (#19677-1-AP, 1:1000), β-actin (#60008-1-lg, 1:1000) (all from Proteintech, Wuhan, China), and cleaved-caspase-9 (#9509, 1:1000), extracellular signal-regulated kinase (ERK) (#12629S, 1:1000), phospho-ERK (Thr202/Tyr204) (#4695S, 1:1000), c-Jun N-terminal kinase (JNK) (#9252, 1:1000), phospho-JNK (Thr183/Tyr185) (#4668, 1:1000), p38 (#8690, 1:1000), and phospho-p38 (Thr180/Tyr182) (#4511, 1:1000) (all from Cell Signaling Technology, Beverly, MA, USA). Membranes were rinsed with TBST, and incubated with appropriate secondary antibodies: horseradish peroxidase (HRP)-conjugated Affinipure goat anti-mouse IgG (H + L) (SA00001-1, 1:10000, Proteintech, Wuhan, China) or HRP-conjugated Affinipure goat anti-rabbit IgG (H + L) (SA00001-1, 1:10000, Proteintech, Wuhan, China) at room temperature for 1 h, then rinsed with TBST and detected using a chemiluminescence detector (Bio-Rad, Hercules, CA, USA) with the Chemiluminescent HRP Substrate (P90719, Millipore, Darmstadt, Germany). The analysis of each protein was performed using Image Lab software (Bio-Rad, Hercules, CA, USA).

### 2.6. Statistical Analysis

Graphpad Prism 7.0 software (Graphpad Software, San Diego, CA, USA) was used for calculating the statistical parameters. Data are expressed as the mean ± standard deviation (SD). Data were analyzed in four independent experiments between the room temperature (RT) vs. cold exposure for 1 week (CE1W) groups, and the cold exposure 1 week (CE2W), and cold exposure 3 weeks (CE3W) groups by one-way analysis of variance. The statistical comparisons and data analyses with *p* < 0.05 were considered as statistically significant.

## 3. Results

### 3.1. The Homeostasis and Neuronal Number of the Hippocampus in Cold-Stressed Mice

To investigate the impact on the homeostasis and the neurons of the hippocampus, the neuron-specific nuclear neuronal expression (NeuN) marker, the dendrite marker, MAP2, and the marker of oxidative stress, malondialdehyde (MDA), were measured in the hippocampus. The MDA levels of hippocampus lysates in the CE1W, CE2W, and CE3W mice were significantly increased when compared with the results of the RT groups (Figure 1). The marker of NeuN and MAP2 were analyzed quantitatively by Western blotting (Figure 2A, B) and immunohistochemistry (Figure 3A). Western blots showed that the NeuN and MAP2 expressions were decreased in the CE1W, CE2W, and CE3W groups (Figure 2C). The results of MAP2 immunohistochemistry of the CA1 and CA3 regions of the hippocampus (Figure 3A) demonstrated that in the CE1W, CE2W, and CE3W groups, the expression of MAP2 in the hippocampal CA1 and CA3 regions were significantly decreased compared to that of the RT group. Proteins in the relevant signaling pathways for anti-oxidative stress in the hippocampal tissues of each group were also investigated using Western blotting (Figure 2B). The results showed an increase in Nrf2 and a decrease in Keap1 (Figure 2D) in the CE1W, CE2W, and CE3W groups compared with that of the RT group. The expression of Nrf2 the signaling pathway proteins superoxide dismutase 1 (SOD1), heme oxygenase (HO-1), and Catalase (CAT) were increased in the cold exposure (CE) groups.

### 3.2. The Effects of Cold Exposure on Neuronal Apoptosis and Proliferation of Cold-Stressed Mice Hippocampus

To confirm the effects of homeostasis imbalance induced by oxidative stress in the hippocampus of cold-stressed mice, the levels of neuronal apoptosis and proliferation were measured by Western blotting, TUNEL staining, and immunofluorescence. The number of apoptotic neurons in the CA1 and CA3 hippocampus of each group was viewed by TUNEL staining (Figure 4A). In the CA1 and CA3 regions, the positive cells of TUNEL (green) staining showed a greater number present in the CE1W, CE2W, and CE3W groups compared to that of the RT group (Figure 4B). The protein level of the endogenous apoptosis-related proteins Cyt-C, Bax, Bcl-2, and executor protein cleaved-caspase-9 and caspase-3 were all measured by Western blotting (Figure 5A). The Cyt-C and the Bax/Bcl-2 ratios were increased in the CE1W, CE2W, and CE3W groups, and the levels of cleaved-caspase-9 and the cleaved/Pro-caspase-3 ratios (Figure 5B) were also significantly increased in the CE1W, CE2W, and CE3W groups following cold stress.

### 3.3. The Response of the MAPK Signaling Pathway in the Hippocampus to Cold Stress

The proteins of the MAPK family signaling pathway were measured to identify the underlying mechanisms of oxidative stress and apoptosis in the chronic cold-stressed mice hippocampus. The expressions of proteins involved in the MAPK signaling pathway components, P-JNK, JNK, ERK, P-ERK, P-p38, p38, and β-actin, were quantitatively determined by Western blotting (Figure 6A). The expressions of JNK Thr183/Tyr185-phosphorylation, ERK Thr202/Tyr204-phosphorylation, and p38 Thr180/Tyr182-phosphorylation levels were all significantly upregulated in the chronic cold-stressed groups, including the CE1W, CE2W, and CE3W groups, compared to that of the RT group (Figure 6B).

## 4. Discussion

In the present study, we showed that neuronal number decreased from chronic cold exposure in the hippocampal CA1 and CA3 regions of mice, and further revealed the mechanism of neuronal apoptosis in cold-stressed mice. All results showed that chronic cold exposure induced cold stress and disrupted the homeostasis of the hippocampus, resulting in oxidative stress and a significant increase in neuronal apoptosis, which significantly affected the integrity of neuronal structures in the hippocampus of mice. Based on the results in the hippocampus, we hypothesized that cold stress impaired hippocampal functions via disrupting the normal homeostasis-induced oxidative stress-mediated MAPK proteins, and the endogenous activated apoptotic pathway increased neuronal apoptosis, finally disrupting the integrity of neurons.

The chronic cold stress model was established based on our previous report [20]. First, we measured the MDA levels in the hippocampus to evaluate the oxidative stress levels. The results indicated that MDA content was significantly enhanced in the CE1W, CE2W, and CE3W groups following cold exposure. MDA is the product of lipid peroxidation of polyunsaturated fatty acids and is used to estimate the lipid peroxidation degree in tissues as a marker for oxidative stress [21]. The levels of MDA indicated that homeostasis was disrupted and oxidative stress was induced. The impact and the response in the hippocampus were next investigated. Based on the results of MDA levels, the key proteins, Nrf2 and Keap1, in the antioxidant signaling pathway were measured. The results indicated that the expression of Nrf2 was significantly increased during cold stress in mice. In contrast, Keap1 was significantly decreased following cold stress. The target genes of Nrf2 SOD1, HO-1, and CAT were all significantly increased, especially SOD1. Nrf2 is a basic leucine zipper protein, exerting an antioxidative effect via regulating the expression of antioxidant proteins to protect against the negative impact of oxidative stress. Under normal physiological conditions, Nrf2 is found in the cytoplasm clustered with the Keap1 protein and is quickly degraded following ubiquitination. Under oxidative stress, Nrf2 is rapidly activated, disrupting the ubiquitination system of Keap1, then translocates to the nucleus, binds to the DNA promoter, and initiates transcription of antioxidative genes such as *SOD* and *HO-1* as a cell survival mechanism [22,23]. Our results showed increased expression of Nrf2 and downstream anti-oxidative proteins, and further suggested the phenomenon of oxidative stress. The markers of neuronal MAP2 and NeuN were measured by Western blotting. The results showed that both marker expressions were significantly decreased in the cold exposure groups (CE1W, CE2W, and CE3W). The expressions were decreased, indicating the neuronal number was affected by oxidative stress following cold stress. In addition, the MAP2 immunohistochemistry results demonstrated that the expression was decreased and the structure of the hippocampus was damaged.

The preliminary results shown above indicated that hippocampal oxidative stress occurred with neuronal loss following cold stress. In order to reveal the relevant mechanism linking neuronal loss with oxidative stress in the hippocampus, the apoptosis level was measured by assessing the endogenous cell death pathway and by using TUNEL staining. TUNEL staining was used to detect DNA fragmentation by labeling the 3′-hydroxyl termini, and it was also used in hippocampal tissue samples to count apoptotic neurons directly [24]. The results showed that the number of TUNEL-positive neurons was significantly increased both in the CA1 and CA3 regions of the hippocampus after exposure to cold stress. The protein expressions and activations relevant to apoptosis were also enhanced. The Western blot results showed that the ratios of Bax/Bcl-2 and cytochrome c (Cyt-c) levels were increased in the hippocampal tissue lysates, and cleaved-caspase 9 and cleaved-caspase 3 expressions were also increased in the cold exposure groups. During apoptosis, Cyt-c is released from the mitochondria when the ratio of Bax/Bcl-2 increases [25]. Bax is a classic pro-apoptotic protein that interacts with the mitochondrial voltage-dependent anion channel, promoting the channel opening and inhibiting the functioning of Bcl-2 then Cyt-c, which mediates caspase-9 activation and cleavage. Once the caspase-9 is activated, cascade amplification and cleaved caspase-3 protein activates apoptosis [26,27]. Our results indicated enhanced apoptosis in the hippocampus of mice exposed to cold stress. Based on our results, the increase in hippocampal apoptosis in mice exposed to cold stress was consistent with the abovementioned phenomenon; neuronal loss via increased apoptosis in the hippocampus following cold exposure was confirmed. The neuronal number was reduced via activation of the intrinsic apoptosis pathway. The neurons of newborn mice level were measured by using Ki-67 immunofluorescence. The Ki-67 protein is a cellular marker for proliferation, and it is associated with cell proliferation [28,29]. The results of Ki-67 immunofluorescence imaging indicated that the newborn neurons were significantly less in the cold exposure groups. This was further confirmed by neuronal number decreases after increased apoptosis, and by inhibition of proliferation because of the homeostatic imbalance following cold exposure.

Finally, the potential mechanism of the homeostatic imbalance following cold exposure was investigated via MAPK signaling pathway activation. It has been reported that MAPK signaling pathway proteins are involved in directing cellular responses to a diverse array of stimuli, such as mitogens, environmental stress, and ultraviolet light exposure [30,31]. The MAPK pathway plays an important role in proliferation, differentiation, cell survival, and apoptosis [32,33]. Our results indicated that members of the MAPK family, including activated/phosphorylated ERK, phosphorylated JNK, and phosphorylated p38 were all upregulated in hippocampus tissue lysates in the cold exposure groups. The results indicated that the MAPK signaling pathway was activated in the hippocampus after homeostasis was disrupted following cold exposure; then, proliferation was inhibited and an increase in apoptosis caused neuronal decreases.

Overall, results indicated that chronic cold exposure induced oxidation stress, demonstrating the impact on the homeostasis of the hippocampus in mice, which increased neuronal apoptosis in the hippocampus during cold exposure. This study provides novel concepts regarding how neurodegenerative diseases may occur; chronic cold stress may be a critical factor to induce neurodegenerative diseases. In the future, we plan to determine how to avoid the risk from chronic cold stress in the hippocampus.

## 5. Conclusions

In conclusion, we have shown that chronic cold exposure induced stress and disrupted the homeostasis of the hippocampus, and mediated oxidation stress, resulting in neuronal number decline. Persistent cold stress-induced oxidative stress and upregulation of the expression of MAPK signaling pathway proteins mediated a downstream intrinsic apoptosis pathway activation, increased neuronal apoptosis, resulting in neuronal decreases and damages to the structures of neurons in the hippocampus.

## Figures and Tables

**Figure 1 brainsci-09-00273-f001:**
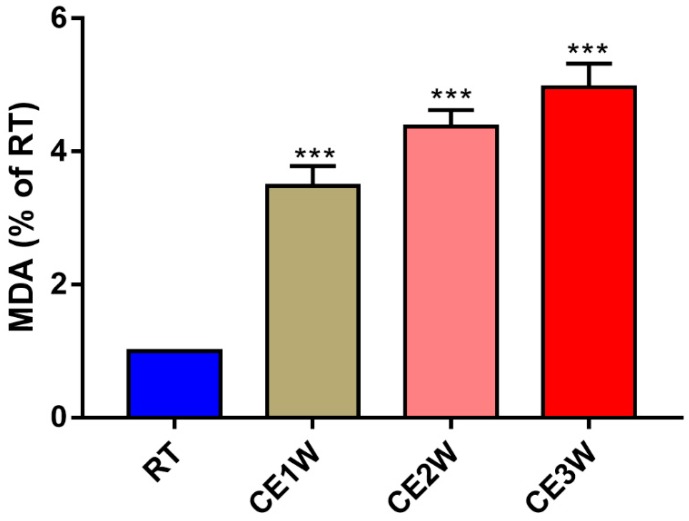
The malondialdehyde (MDA) level in hippocampal lysates of the chronic cold-stressed mice were measured. Data were between room temperature male (RTM) vs. cold exposure for 1 week (CE1W), CE2W and CE3W groups and were analyzed by one-way analysis of variance. The results are expressed as the mean ± SD (*n* = 6) of four independent groups; *** *p* < 0.001.

**Figure 2 brainsci-09-00273-f002:**
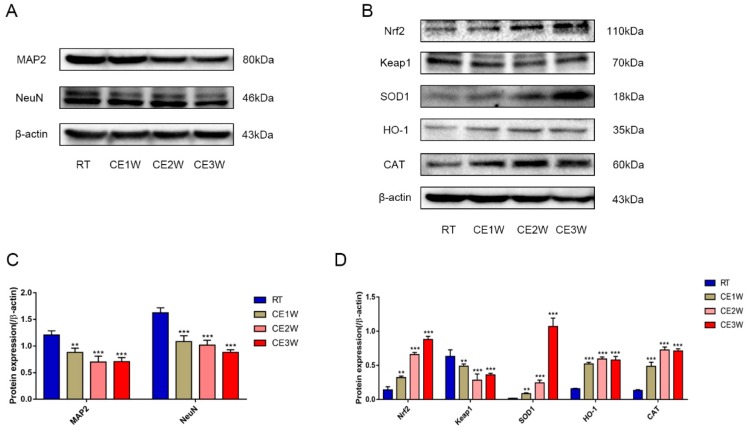
The function and target protein expression levels of anti-oxidative proteins in the anti-oxidative signaling pathway and markers of neuron-specific nuclear and dendritic tissues in hippocampal lysates as assessed by Western blotting and relative density analyses. (**A**) MAP2 and neuronal expression (NeuN). (**B**) Nrf2, Keap1, superoxide dismutase 1 (SOD1), heme oxygenase (HO-1), and Catalase (CAT) and β-actin (control) expressions in each group. The expression levels of each protein was quantitated by measuring the band intensities using Image Lab software. The graphs indicate densitometric analyses using the expression ratios of (**C**) MAP2 and NeuN/β-actin, and (**D**) Nrf2, Keap1, SOD1, HO-1, and CAT/β-actin. Data were between RTM vs. CE1W, CE2W and CE3W groups and were analyzed by one-way analysis of variance. The results are expressed as the mean ± SD (*n* = 6) of four independent groups. ** *p* < 0.01; *** *p* < 0.001.

**Figure 3 brainsci-09-00273-f003:**
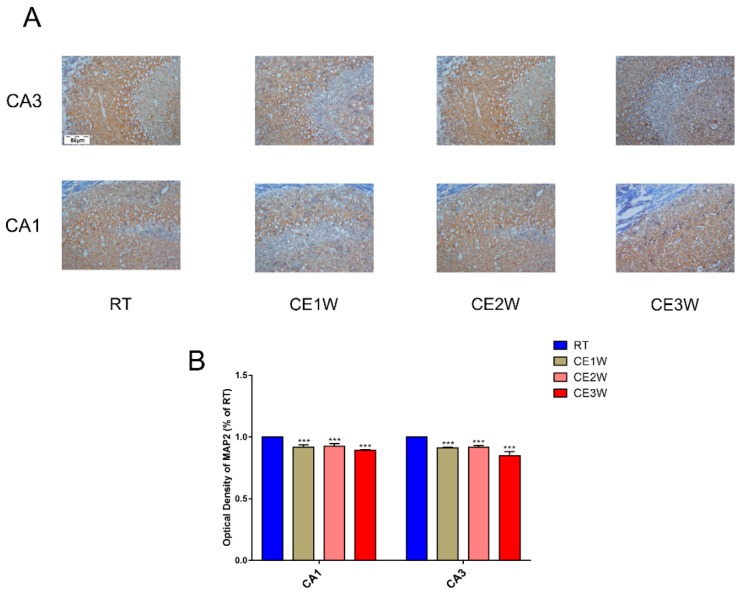
The marker expressions of neuronal dendrites in the hippocampus. MAP2 was measured by immunohistochemistry in the CA1 and CA3 regions of cold-stressed mice. (**A**) The positive staining in the CA1 and CA3 regions of the hippocampus showing MAP2 expression by immunohistochemical in each group. Scale bar = 86 μm. The graphs (**B**) demonstrate the results of MAP2 optical density in the functional regions of the hippocampus CA1 and CA3 regions. Data were between RTM vs. CE1W, CE2W and CE3W groups and were analyzed by one-way analysis of variance. The results are expressed as the mean ± SD (*n* = 3) of four independent groups. *** *p* < 0.001.

**Figure 4 brainsci-09-00273-f004:**
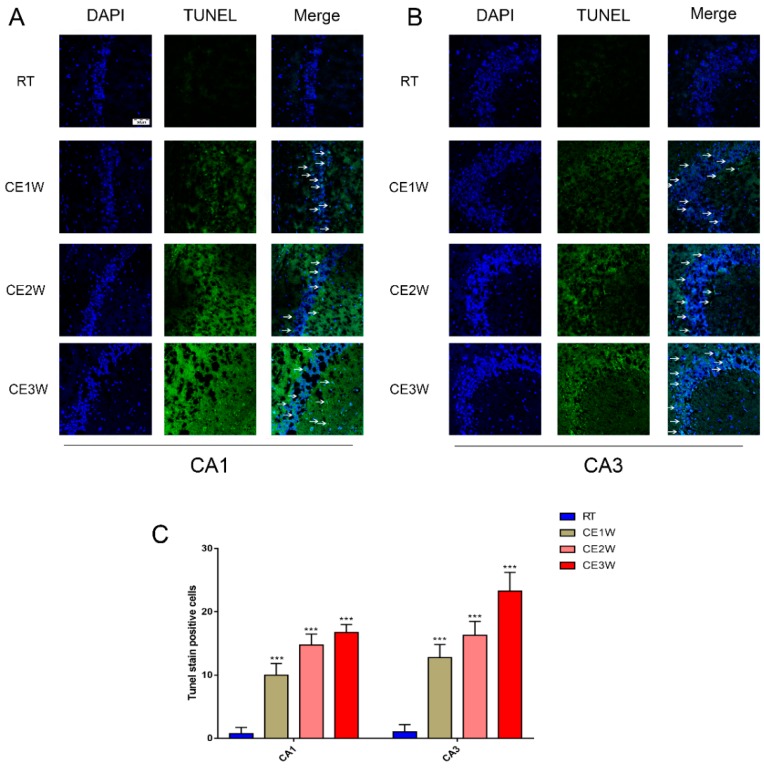
The number of apoptosis cells in the CA1 and CA3 hippocampus regions of each group was directly viewed and counted by TdT-mediated dUTP Nick-End Labeling (TUNEL) staining. The results of TUNEL staining indicated TUNEL (green) staining positive cells counterstained with 4’,6-diamidino-2-phenylindole (DAPI) (blue) in the CA1 and CA3 regions (**B**) of the hippocampus (**A**) in each group. Scale bar = 50 μm. The apoptotic cells were counted and showed that the number of TUNEL-positive cell in hippocampal CA1 and CA3 regions (**C**) of each group. Data were between RTM vs. CE1W, CE2W, and CE3W groups and were analyzed by one-way analysis of variance. The results are expressed as the mean ± SD (*n* = 3) of four independent groups. *** *p* < 0.001.

**Figure 5 brainsci-09-00273-f005:**
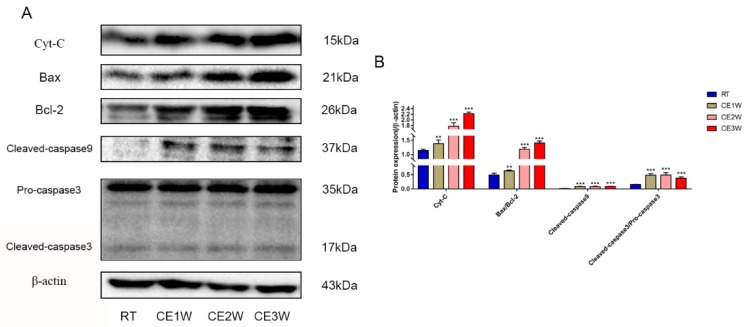
The expressions of kinases and key proteins on mitochondrial apoptosis signaling pathway. (**A**) Cyt-c, Bax, Bcl-2, cleaved-caspase-9, caspase-3, and β-actin (control) expressions in the hippocampal as assessed by Western blotting; the relevant expressions were analyzed and shown as graphs (**B**) in each group. Data were between RTM vs. CE1W, CE2W, and CE3W groups and were analyzed by one-way analysis of variance. The results are expressed as the mean ± SD (*n* = 6) of four independent groups. ** *p* < 0.01; *** *p* < 0.001.

**Figure 6 brainsci-09-00273-f006:**
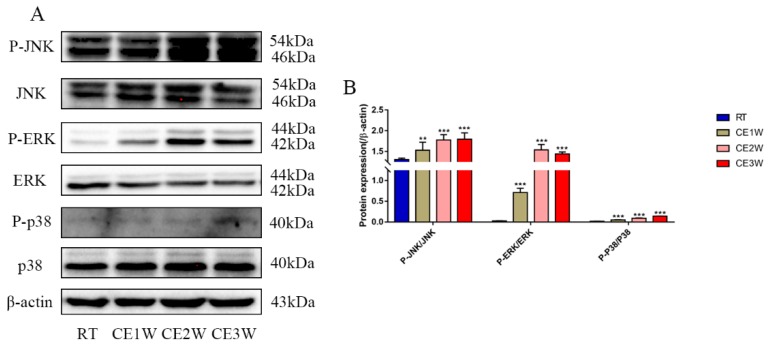
The expressions of MAPK signaling pathway components in hippocampal lysates as assessed by Western blotting and relative density analyses. (**A**) JNK, phospho-(P-) JNK (Thr183/Tyr185), ERK, P-ERK (Thr202/Tyr204), p38, P-p38 (Thr180/Tyr182), and β-actin (control) expressions. The relevant kinase expressions were analyzed and shown as graphs (**B**) in each group. Data were between RTM vs. CE1W, CE2W, and CE3W groups and were analyzed by one-way analysis of variance. The results are expressed as the mean ± SD (*n* = 6) of four independent groups. ** *p* < 0.01; *** *p* < 0.001.

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
