# Peer review of "Oxidation Stress-Mediated MAPK Signaling Pathway Activation Induces Neuronal Loss in the CA1 and CA3 Regions of the Hippocampus of Mice Following Chronic Cold Exposure"

_brainsci, 2019, doi:10.3390/brainsci9100273_

Round 1
Reviewer 1 Report
This is a review of the manuscript “Oxidation stress-mediated MAPK signaling pathway activation induces neuronal loss in the CA1 and CA3 regions hippocampus of mice following chronic cold exposure” by Bin Xu, et.al., from Heilongjiang Bayi Agricultural University, Daqing, China. C57BL/6 mice were exposed to chronic cold stress involving 40C, 3 hours per day for 1, 2, or 3 weeks. Homeostasis in the hippocampus, neuronal apoptosis, and neuronal proliferation were evaluated by western blotting, immunohistochemistry, TUNEL staining, and immunofluorescence. The phenomena of oxidation stress, MAPK signaling pathway activation, anti-oxidation protein release, neuronal apoptosis increases, and neuronal proliferation inhibition were demonstrated in mouse CA1 and CA3 regions of hippocampal tissues following cold exposure.
Overall, the manuscript has merit and demonstrates interesting changes that have occurred in the hippocampus following the chronic cold stress. However, there are numerous parts of this manuscript that need clarification or reworking. The following is a list of details to attend to:
In the experimental design, what gender of mice have been used? Line 15 in the Abstract says the animals are all male mice. Line 62 in section 2.1 says that both male and female mice were used. If both genders of mice were utilized, explain the rationale for that choice. If both genders of mice were utilized, list the number of male and female animals in each of the 4 groups. Clarify how many animals were in each group. The current description is confusing. Line 76 says n = 3 per group, and line 82 says n = 6 per group. Clarify what numbers were utilized for the statistical analyses. In numerous places of the manuscript, it is written that, “The results were analyzed in four independent experiments, the data are expressed as the mean + SD.”(see lines 157, 166, 173, 197, 203, 211, 225). Does this mean that each of these experiments was run 4 different times? If so, what numbers were utilized to express the mean, was it the mean of the 4 experiments, and what numbers were utilized for the statistical analysis? In Section 2.7. Statistical analysis, there is no mention of any post hoc tests to determine differences between the experimental groups. Thus the following statements need to be deleted: lines 151-152, “and both proteins changed in a time-dependent manner”; lines 153-154, “and all proteins were increased in time-dependent of cold exposure”; lines 240-241, “groups in a time-dependent manner”; line 247, “in a time-dependent manner” The quality of the data in Figure 6 seems very questionable and in the opinion of this reviewer, should not be used in this manuscript. The images presented in Figure 6A are not clear in what the authors are wanting to demonstrate. All of the image looks blue and thus the specific Ki67 staining is very subjective and very questionable. I do not trust that data based on the images presented in Figure 6A. The data about Ki67 needs to be removed from the manuscript. The writing in some of the sections seems fairly adequate, but in other sections there are numerous typos and grammatical errors in the writings. All of the manuscript needs to have appropriate editorial scrutiny. Sections 2.1, 2.2, 2.3, 2.4, 2.5, 2.6, 2.7, and 3.1 especially need this scrutiny. Figure legends 1-7 need to change to “The results were analyzed…”. Also check typos on lines 110, 133, 145, 207, 230, and 280. On line 4 of the title, add “of the” between region and hippocampus. The graph in Figure 5B is much too small to be able to be read. This needs to be enlarged, or broken into two graphs to allow it to be enlarged. The Discussion needs to have more references. For all of the information given in the Discussion, there needs to be more back up from the literature about this background information. Should lines 316-319 be removed?
Reviewer 2 Report
Lang et al., has presented a study investigating the mechanisms underlying potential oxidative stress and neuronal loss in the hippocampus in mice exposed to cold temperatures. The authors showed that mice exposed to 4-degree temperatures for 4 hours a day, for various weeks, had increased oxidative stress in a time dependent manner. This oxidative stress is thought to be mediated through the Nrf2 pathway and leads to increase downstream antioxidants such as HO-1, SOD and CAT. The authors also noted that increased exposure to cold temperatures reduced MAP2 neuronal staining and increased TUNEL staining in the CA1 and CA3 regions of the hippocampus, suggesting a loss in the neuronal population with cold exposure. This neuronal loss was accompanied by reduced cell proliferation in the CA1 and CA3 and increase expression of apoptotic and ERK signaling pathway proteins in the hippocampus. Given this data, the authors have presented a nice study showing that increased exposure to cold temperatures can lead to neuronal loss and increased oxidative stress in the hippocampus of mice.
The following minor revisions should be considered.
The text contains grammatical that need to be revised. Some pictorial representations of immunohistochemistry and western blots do not match the reported quantitative data. In the MAP2 staining of the CA1 and CA3 regions, using images of similar regions in these areas will be better for comparisons, however these changes do not seem to be highly significant. Further analysis needs to be performed to confirm changes of MAP2 in the CA1 and CA3 regions of the hippocampus. Lastly, in the Ki67 pictures, it is difficult to see true changes in the staining, even with the white arrows. Consider a new way of highlighting these puncta. The p-p38 data is not convincing. Authors should rerun these experiments over using a different p-p38 antibody or different western blotting protocol to prove that p-p38 is increased during cold exposure. The author has reported statistical analysis on each timepoint compared to room temperature exposed mice. However, in the reporting, the author suggests that there is a time dependent difference in many of these experiments. Authors should perform the proper statistical test to determine if there are significant changes amongst time points.
